# How does domestic migration pose a challenge in achieving equitable social health insurance benefits in China? A national cross-sectional study

Haiqin Wang ,[1,2] Di Liang,[2] Donglan Zhang,[3] Zhiyuan Hou [2]

[1]Administrative Office, The International Peace Maternity and Child Health Hospital of China Welfare Institute, Shanghai, China
[2]School of Public Health, NHC Key Laboratory of Health Technology Assessment, Fudan University, Shanghai, China
[3]Division of Health Services Research, Department of Foundations of Medicine, New York University Long Island School of Medicine, Mineola, New York, USA

**Correspondence to**
Dr Zhiyuan Hou;
zyhou@fudan.edu.cn

## ABSTRACT

**Objectives** To evaluate the benefit distribution of social health insurance among domestic migrants in China.

**Design** A national cross-sectional survey.

**Setting** 348 cities from 32 provincial units in China.

**Participants** 1165 domestic migrants who used inpatient care services in the city of a new residence and had social health insurance.

**Primary and secondary outcome measures** The probability of receiving reimbursements from social health insurance, the amounts and ratio of reimbursement received.

**Results** Among migrants who used inpatient care in 2013, only 67% received reimbursements from social health insurance, and the reimbursement amount only accounted for 47% of the inpatient care expenditure. The broader the geographical scope of migration, the lower the probability of receiving reimbursement and the reimbursement ratio, but the higher the reimbursement amount. Specifically, the probability of receiving reimbursements for those who migrated across cities or provinces was significantly lower by 14.7% or 26.0%, respectively, than those who migrated within a city. However, they received significantly higher reimbursement amounts by 33.4% or 27.2%, respectively, than those who migrated within a city. And those who migrated across provinces had the lowest reimbursement ratio, although not reaching significance level.

**Conclusions** The unequal benefit distribution among domestic migrants may be attributed to the fragmented health insurance design that relies on localised administration, and later reimbursement approach that migrating patients pay for health services up-front and get reimbursement later from health insurance. To improve the equity in social health insurance benefits, China has been promoting the portability of social health insurance, immediate reimbursement for inpatient care used across regions, and a more integrated health insurance system. Efforts should also be made to control inflation of healthcare expenditures and prevent inverse government subsidies from out-migration regions to in-migration regions. This study has policy implications for China and other low/middle-income countries that experience rapid urbanisation and domestic migration.

## STRENGTHS AND LIMITATIONS OF THIS STUDY

⇒ Data are from a 2014 nationally representative cross-sectional sample of domestic migrants in China eight years ago, and a two-part model is used.
⇒ Sensitivity analyses are conducted on a total sample and two subsamples to get robust results.
⇒ The outcomes related to insurance reimbursement were self-reported, leading to measurement bias.
⇒ Health status may influence healthcare utilisation and migration status.

## INTRODUCTION

Over the past decades, China has been experiencing rapid urbanisation and internal migration.[1] Domestic (internal) migrants who lived outside their place of origin reached 375 million in 2020.[2] However, migrants faced huge barriers to obtaining health insurance where they lived.[3 4] Research has shown that migrants were less likely to be covered by social health insurance than permanent residents.[4 5] Although China established a national social health insurance system and made efforts to make the social health insurance system portable, this system was quite fragmented and administrated locally, leading to considerable variation in insurance benefits across health insurance programmes and regions.[6] The unequal benefits in insurance might be amplified among the migrant population for the following reasons.

First, the social health insurance system consists of three separate insurance programmes designed based on citizen's residence registration status and employment status[5]: the New Rural Cooperative Medical Scheme (NCMS) covering the registered rural residents, the Urban Employee Basic Medical Insurance (UEBMI) covering urban employees only and the Urban Resident Basic Medical Insurance (URBMI) covering urban non-working residents.[6–8] UEBMI is jointly

financed by employers and employees, while URBMI and NCMS are chiefly financed by general taxes and individual premiums. Since 2016, integrating URBMI and NCMS into Urban–Rural Resident Basic Medical Insurance has been piloted in some provinces and gradually expanded nationally, although it is still administrated locally.[9] Subsidies from the government account for up to 70% of the URBMI and NCMS funds. Among the three insurance programmes, UEBMI offers the most generous benefits package that provides reimbursement for both outpatient and inpatient services at high rates. URBMI and NCMS mainly reimburse for inpatient services.[10] To promote urbanisation, the Chinese government issued policies to allow migrants to enrol in UEBMI or URBMI depending on their employment status.[11] Thus, migrants may be enrolled in any of the three insurance programmes, which may result in the variation in insurance benefits among migrants.

Second, the social health insurance programmes are administrated and financed by the local county or city government.[12] Each county or city designs its benefits package, which mainly covers health services delivered within the county/city and generally does not reimburse health services delivered outside the county/city.[10] Even if some counties/cities cover health services outside the county/city, the lack of insurance portability across regions could create an additional barrier for insurance enrollees to receive benefits. Migrants may enrol in health insurance at one place but move to another and receive health services in this new place. They need to go back to the place where they are enrolled to get insurance reimbursement. To address the insurance reimbursement issue for migrants, China promoted immediate reimbursement for cross-province medical care in 2014.[13] However, the immediate reimbursement policy was only used for inpatient services, and immediate reimbursement for outpatient services was piloted in a few regions in 2020.[14] Despite these policies, the separation between the location of healthcare use and health insurance coverage are still barriers to equity benefits.[15]

Previous benefit analyses among the general population show that government subsidies for social health insurance were pro-rich for inpatient and outpatient services from 2003 to 2013 in China, although the inequity in benefit distribution had been narrowed.[16–18] Evidences from URBMI (between 2007 and 2011)[19] and the social health insurance (from 2014 to 2016)[20] reveal that the lower-income groups benefited less than the higher-income groups. The poorest groups within URBMI and NCMS were consistently more likely to forego hospitalisation services recommended by doctors than their wealthier counterparts between 2008 and 2018.[9] With the fragmented feature and increased benefit disparities, social health insurance reinforces the existing rural–urban inequity and generate a new inequity between urban residents and migrants who lived in urban areas.[8] In recent years, more literature document barriers to preventing migrants from gaining access to healthcare[21 22]

and health insurance had little influence on healthcare utilisation among internal migrants.[22–24] A study by Li et al further concludes that NCMS did not play a significant role in reducing out-of-pocket payments for elderly migrants between 2005 and 2014.[25] In addition to these disparities, there is a unique contributor to health inequity among migrants—the scope of migration—where they migrated to. However, no literature has focused on the benefit distribution of social health insurance by the scope of migration for the enormous migrant population in China.

Using the 2014 China National Internal Migrants Dynamic Monitoring Survey, we conduct a benefit analysis of social health insurance among a representative sample of migrants in China. This study is the first to assess the benefit distribution by the geographical scope of migration and health insurance programmes. We aim to generate new evidence on China's continuously changing health insurance system and provide policy implications for other developing countries striving to achieve universal health coverage under rapid urbanisation.

## METHODS
### Data and study design
Data used in this analysis are from the 2014 China National Internal Migrant Dynamic Monitoring Survey. The survey was conducted by the National Health and Family Planning Commission of China in May 2014. This was a national cross-sectional survey representing internal migrants aged 15–59 years who had lived in a city of a new residence for more than 1 month but did not have a 'Hukou' of the city (registered resident certificate).

In this survey, a stratified multistage random sampling method by probability proportional to size (PPS) was employed. The annual national data on internal migrants from each province in 2013 was considered as the primary sampling frame. A total of 348 cities from 32 provincial units in China were surveyed. Within each city, townships were randomly selected and followed by neighbourhoods using the PPS. And then, in each neighbourhood, 20 internal migrants were randomly selected to participate in the survey, finally reaching a total sample of 200 937 respondents. The face-to-face interview was conducted by trained interviewers using a structured questionnaire. Informed consent was sought from the study respondents.

Questionnaires include demographic information, family structures, socioeconomic status, migration characteristics, health insurance, healthcare services and family planning services. In this study, we focus on internal migrants who used inpatient care in the city of a new residence during the previous year of the survey and analyse the benefits they received from social health insurance. Thus, our sample comprises internal migrants who used inpatient care services in the city of a new residence and had social health insurance, with a sample size of 1165.

## Measurements

In this analysis, health insurance benefits are measured by three outcomes: the probability of receiving reimbursements from social health insurance, the total amount of reimbursement received and the percentage of reimbursements of total healthcare expenditures (reimbursement ratio). In the survey, we identify the first outcome by a multiple-choice question, 'where did you receive reimbursement for your last hospitalization this year.' Answers include: allowance from NCMS, allowance from UEBMI, the employer, the NCMS office, the local health centres, the commercial insurance, allowance from the Family planning operation, the family planning operation and else. We recognise those who answered only 'the commercial insurance' or 'else' or both 'the commercial insurance' and 'else' as receiving no reimbursement from social health insurance. We identify the second outcome by the question 'how much reimbursement did you receive from social health insurance.' We identify the last outcome by the question for the second outcome and the question 'how much did you cost in total.'

Our primary predictors of interest are social health insurance programmes and the geographical scope of migration. Social health insurance programmes include UEBMI, URBMI and NCMS. Since the social health insurance programmes are administrated and financed by county or city, we hypothesise that there are benefit disparities for migrants rooted in the separate administration of insurance programmes. Therefore, we categorise migrants according to their scope of migration, which can capture the degree of separate administration of insurance programmes. The geographical scope of migration is categorised into three subgroups: migration across counties within a city, migration across cities within a province and migration across provinces (Under China's administrative division, a county is smaller than a city).

Controlled variables include demographic characteristics, socioeconomic status, other migration characteristics and the facility level for hospitalisation. Demographic characteristics include gender, age and marital status. Marital status is measured by a binary variable indicating whether the respondent was married or not married (eg, widowed, divorced or never married). Socioeconomic status is measured by educational attainment, monthly household income per capita, whether the respondent had a job, whether the respondent had rural 'Hukou,' and whether the respondent lived in urban areas. Educational attainment is categorised into four subgroups: primary school and below, junior high school, senior high school, and college degree and above. 'Hukou' represents the record in the residency registration system in China; people can be registered as having either a rural or urban 'Hukou' at birth and cannot be easily changed throughout their lifetime.[26] Other migration characteristics are measured by migration reasons and duration. The reasons for migration include seeking jobs, family members following them to migrate, or other reasons. Migration duration is categorised into four groups: less

than 1 year; 1– years; 5– years; and 10 years and above. Finally, the hospitalisation facility-levels include primary care facility, secondary hospital, tertiary hospital and private hospital.

## Statistical analysis

We first describe the general characteristics of our study sample. $\chi^2$ test and one-way variance analysis are used to compare the differences in the probability of receiving reimbursement and the amount and ratio of the reimbursement according to the geographical scope of migration.

Since there are many 'zero observations'—patients who used inpatient care but received no reimbursement, we use the two-part model to estimate the benefits migrants received from the social health insurance, which can be expressed as follows:

$$\Pr\left[(Reimburse\ amount)_i > 0\right] = \Phi\left[\beta_1 * (Migration\ scope)_i + \eta' * X_i + \varepsilon_i\right]$$
(1)

$$\log\left[(Reimburse\ amount)_i \mid (Reimburse\ amount)_i > 0\right] = \beta_2 * (Migration\ scope)_i + \eta' * X_i + \varepsilon_i$$
(2)

$$(Reimburse\ ratio)_i \mid (Reimburse\ amount)_i > 0 = \beta_3 * (Migration\ scope)_i + \eta' * X_i + \varepsilon_i$$
(3)

Where $(Reimburse\ amount)_i$ and $(Reimburse\ ratio)_i$ are the reimbursement amount and ratio received by individual i. $(Migration\ scope)_i$ is a set of dummies representing migration scopes of individual i, and migration across counties within a city is taken as the reference group. The parameters $\beta_1$, $\beta_2$ and $\beta_3$, the key coefficients of interest, identify the association between migration scope and the probability of receiving reimbursement, and the amount and ratio of the reimbursement conditional on reimbursement received, respectively. $X_i$ is a vector of control variables, including social health insurance coverage, demographic characteristics, socioeconomic status, other migration characteristics and the facility level for hospitalisation.

The above two-part model assumes that the benefits migrants received from social health insurance were determined by two separate decision-making processes: equation (1), the 'participation equation,' captures the fundamental difference between the respondents who received reimbursements from social health insurance and those who did not; as the 'intensity equation,' equation (2) and equation (3) characterise the determinants of the amount and ratio of the reimbursement received among those who received reimbursements. In equation (2) and equation (3), the amount and ratio of the reimbursement fit the Gamma distribution, and the logarithm transformation is taken on the reimbursement amount to reduce the impact of extreme values. Following the previous study (eg, Jan pan, Sen Tian, Qin Zhou and Wei Han) (2016),[19] we estimate equation (1) with the probit model, and equation (2) and equation (3) with the

generalised linear model, respectively. Marginal effects with SEs are reported.

All the analyses are conducted for the total sample, rural social health insurance sample (the NCMS subsample), and urban social health insurance sample (the URBMI and UEBMI subsample), respectively. All analyses are performed using STATA V.12.0 (StataCorp).

### Patient and public involvement

No patients were actively involved in setting the research question, outcome measures nor involved in the design of the study. Patients were not involved in interpretation or write up of the results, nor are there plans for the results to be disseminated to the patient community affected by this research.

## RESULTS
### Characteristics of the study sample

Table 1 presents the descriptive statistics for our study sample. Of the 1165 respondents, 66.70% enrolled in NCMS and 23.00% and 10.30% enrolled in UEBMI and URBMI, respectively. The average expenditures per inpatient stay were ¥10 366 (=US$1535), and there was a small difference (about ¥500, p=0.643) in inpatient expenditure between the NCMS and URBMI and UEBMI subsample. 66.78% of respondents who used inpatient care received reimbursement from social health insurance, with 60.49% for NCMS enrollees and 79.38% for URBMI and UEBMI enrollees. Among the respondents who received reimbursement, the average amount and ratio of the reimbursement received were ¥5506 (=US$815) and 46.77%. The average amount and ratio of the reimbursement received for NCMS enrollees were much smaller than those for URBMI and UEBMI enrollees.

The average age and monthly household income per capita of the respondents were 38 years old and ¥2256. Less than half of the respondents were female. Most of them were married (89.27%), had an education level of high school or below (89.70%), had rural 'Hukou' (87.81%), owned a job (79.57%) and lived in urban areas (69.36%). Nearly half of the respondents migrated across provinces, while those who migrated across cities but within a province and those who migrated across counties but within a city were 28.76% and 25.41%, respectively. 84.21% of respondents migrated for better job opportunities, and 87.38% of respondents had lived in the city of a new residence for more than 1 year. Most respondents (80.51%) chose inpatient care at secondary and tertiary hospitals instead of primary care facilities.

We compare the characteristics of the study sample by the scope of migration (online supplemental appendix table 1). The demographic and socioeconomic characteristics of migrants are similar across three scopes of migration, including age, sex, marriage status, education, job status, Hukou status, reasons for migration, migration duration and health insurance programmes they enrolled in. The Only exception is income and living

areas. Migrants across provinces had more income and were less likely to live in urban areas than other groups.

Table 2 summarises the total expenditures per inpatient stay and the probability of benefiting from social health insurance. It also presents the reimbursement amount and ratio received among the benefit recipients according to the geographical scope of migration. The univariate analyses show that the broader the migration scope, the lower the probability that migrants would receive reimbursements; but among those who received reimbursements, those who migrated across cities or provinces received larger amounts of reimbursement than those who migrated within a city. There is no significant difference in total expenditure and reimbursement ratio by the geographical scope of migration.

### Association between insurance programmes, migration scope and social health insurance benefits

Table 3 reports the association between insurance programmes, migration scope, other factors and social health insurance benefits, estimated from a two-part model. Compared with NCMS enrollees, URBMI or UEBMI enrollees were more likely to receive reimbursement, and among the benefit recipients, urban insurance enrollees received a larger reimbursement amount and ratio. The probability of receiving reimbursement for UEBMI enrollees was 37.5% (p<0.01) higher than that for the NCMS enrollees. Among insurance benefit recipients, UEBMI enrollees received 42.8% (p<0.01) more reimbursement amount and 20.1% (p<0.01) higher reimbursement ratio than NCMS enrollees.

According to the association between insurance benefits and migration scope, the geographical scope of migration reduced the probability of receiving reimbursement and the reimbursement ratio but increased the reimbursement amounts they received. Specifically, the probability of receiving reimbursement for those who migrated across cities or provinces was significantly lower by 14.7% or 26.0%, respectively, than those who migrated within a city (p<0.01). However, they received significantly higher reimbursement amounts by 33.4% or 27.2%, respectively, than those who migrated within a city (p<0.01). And those who migrated across provinces had the lowest reimbursement ratio, although not reaching significance level.

In addition, there is no significant difference in insurance benefits by age, gender, marriage status, education, Hukou status, and migration duration. Income had no significant influence on the probability and ratio of reimbursement but significantly increased the reimbursement amount. Having jobs significantly decreased the reimbursement amount, whereas living in urban areas increased the probability of receiving reimbursement by 6.3% compared with living in suburban areas. Compared with migration for seeking jobs, family members following migrants significantly increased the reimbursement amount. The higher the level of healthcare facility, the greater the reimbursement amount, but the lower the reimbursement ratio.

**Table 1** Characteristic of the study sample, n (%)

| Variables | Total sample (N=1165) | NCMS subsample (N=777) | URBMI and UEBMI subsample (N=388) |
|---|---|---|---|
| Total expenditure per inpatient stay (Yuan)* | 10366.05 (±17549.73) | 10197.45 (±18374.85) | 10704.99 (±15778.28) |
| Probability of receiving reimbursement | 778 (66.78) | 470 (60.49) | 308 (79.38) |
| Reimbursement amount received (Yuan)* | 5506.16 (±9761.79) | 5128.44 (±10838.10) | 6049.14 (±7952.00) |
| Reimbursement ratio received (%)* | 46.77 (±19.91) | 39.41 (±17.86) | 57.35 (±17.87) |
| Social health insurance programmes | | | |
| NCMS | 777 (66.70) | – | – |
| URBMI | 120 (10.30) | – | 120 (30.93) |
| UEBMI | 268 (23.00) | – | 268 (69.07) |
| Age (years)* | 37.65 (±9.75) | 37.79 (±9.80) | 37.37 (±9.65) |
| Female | 532 (45.67) | 361 (46.46) | 171 (44.07) |
| Married | 1040 (89.27) | 705 (90.73) | 335 (86.34) |
| Education attainment | | | |
| Primary school and below | 279 (23.95) | 218 (28.06) | 61 (15.72) |
| Junior high school | 554 (47.55) | 416 (53.54) | 138 (35.57) |
| Senior high school | 212 (18.20) | 118 (15.19) | 94 (24.23) |
| College and above | 120 (10.30) | 25 (3.22) | 95 (24.48) |
| Monthly household income per capita (Yuan)* | 2256.43 (±2092.85) | 1999.55 (±1687.47) | 2770.86 (±2658.26) |
| Having any job | 927 (79.57) | 610 (78.51) | 317 (81.70) |
| Rural Hukou | 1023 (87.81) | 777 (100.00) | 267 (68.81) |
| Living in an urban area | 808 (69.36) | 502 (64.61) | 306 (78.87) |
| Geographical scope of migration | | | |
| Across counties within a city | 296 (25.41) | 216 (27.80) | 80 (20.62) |
| Across cities within a province | 335 (28.76) | 214 (27.54) | 121 (31.19) |
| Across provinces | 534 (45.84) | 347 (44.66) | 187 (48.20) |
| Reasons for migration | | | |
| Seeking jobs | 981 (84.21) | 642 (82.63) | 339 (87.37) |
| Family members following migrants | 146 (12.53) | 115 (14.80) | 31 (7.99) |
| Other reasons | 38 (3.26) | 20 (2.57) | 18 (4.64) |
| Migration duration (years) | | | |
| 0– | 147 (12.62) | 117 (15.06) | 30 (7.73) |
| 1– | 484 (41.55) | 305 (39.25) | 179 (46.13) |
| 5– | 272 (23.35) | 181 (23.29) | 91 (23.45) |
| 10+ | 262 (22.49) | 174 (22.39) | 88 (22.68) |
| Facility level of hospitalisation | | | |
| Primary care facility | 131 (11.24) | 90 (11.58) | 41 (10.57) |
| Secondary hospital | 400 (34.33) | 282 (36.29) | 118 (30.41) |
| Tertiary hospital | 538 (46.18) | 340 (43.76) | 198 (51.03) |
| Private hospital | 96 (8.24) | 65 (8.37) | 31 (7.99) |

*Mean (±SD).
NCMS, New Rural Cooperative Medical Scheme; UEBMI, Urban Employee Basic Medical Insurance; URBMI, Urban Resident Basic Medical Insurance.

Considering the differences in reimbursement policy between NCMS and urban health insurance, we further conduct the above regressions among the subsamples of NCMS enrollees and URBMI and UEMBI enrollees (table 4). The relationships between migration scope and the reimbursement probability and amount did not change, but they differed with the reimbursement ratio. Those who migrated more broadly had a significantly lower reimbursement ratio among NCMS enrollees, while that among URBMI and UEMBI enrollees was higher.

**Table 2** Benefit distribution of social health insurance by geographical scope of migration

| Variables | Across counties within a city | Across cities within a province | Across provinces | P value |
|---|---|---|---|---|
| Total expenditure per inpatient stay (Yuan) | 10 326.81 (14 079.35) | 14 458.96 (27 139.10) | 12 771.68 (20 428.78) | 0.354 |
| Probability of receiving reimbursement (%) | 81.76 (38.69) | 69.85 (45.96) | 56.55 (49.62) | <0.001 |
| Reimbursement amount received (Yuan) | 4402.12 (6541.01) | 6706.32 (13 681.56) | 5495.89 (8105.80) | 0.056 |
| Reimbursement ratio received (%) | 44.75 (18.17) | 47.76 (20.51) | 47.72 (20.79) | 0.195 |

Note: SD are reported in parentheses.

## Reasons for not receiving reimbursement from social health insurance

We further investigate why migrants did not receive reimbursement from their social health insurance. Figure 1 shows that the need or plan to go back to hometown to get reimbursement was the main reason for not getting reimbursement, accounting for 66.5%, followed by a lack of knowledge about the reimbursement process (16.7%) and the policy coverage issues (10.6%). Figure 2 compares the proportion of not receiving reimbursement due to the need or plan to go back to their hometown by migration scope. The broader the migration scope, the higher the likelihood that migrants did not receive reimbursement because they must get it later from their hometowns.

## DISCUSSION

We find that the probability of receiving reimbursement and the reimbursement ratio for migrants were far lower than those for the general population. Only 60% of migrants who were NCMS enrollees received reimbursement from NCMS, which was 30% lower than that of general NCMS enrollees (91.1% in 2013), and the reimbursement ratio was 10% lower than that of general NCMS enrollees (39.4% vs 50.1% in 2013).[27] For URBMI and UEBMI enrollees, the probability of receiving reimbursement among migrants was about 10% lower than that of the general population.[27] A previous study also points out that migrants only partially benefited from health insurance coverage.[24]

Our findings should be understood in the China-specific context. In China, migrants face more challenges in obtaining insurance reimbursement than non-migrants. There have been two common approaches to reimbursing healthcare services (immediate reimbursement and later reimbursement).[28] Immediate reimbursement means that the insured patients get reimbursement immediately for the treatment and only pay out-of-pocket for the copay or coinsurance rate, whereas later reimbursement means that the insured patients pay for the total expenditures out-of-pocket up-front and get reimbursement later from their health insurance.[28] Residents usually get reimbursement immediately, but migrants generally receive reimbursement later as they must travel back to their hometown where they enrol in

social health insurance. Our study shows that up to 22% of migrants reported that they did not get reimbursement because they needed to go back to their hometowns. In addition, research has shown that many services were not reimbursed, and the reimbursement process for migrants was much more complex than that for residents.[29] One study shows that compared with residents, more migrants were treated in hospitals outside the NCMS designated network, and thus their healthcare uses were less likely to be covered by NCMS.[29]

Our main findings show inequity in insurance benefits among migrants by migration scope. Although the scope of migration was associated with a larger reimbursement amount per inpatient stay, it was significantly associated with a lower probability of receiving reimbursement. All three social health insurance programmes in China were administered, financed and operated by local county or city governments. Each county or city designed its benefits packages and made the benefit localised,[10 30] limiting individual coverage choices outside the local region. This poses a challenge for internal migrants who typically use healthcare in the city of new residence but may enrol in health insurance at their hometown according to their residence ('Hukou') status. The separation between where healthcare was received and where health insurance was administrated provided an additional hurdle for the internal migrants to receive benefits from their social health insurance. It became even more difficult when they lived far away from their hometowns. The localised administration of social health insurance and the later reimbursement approach contributed jointly to the inequity benefit for migrants.[31 32] To resolve this inequity, China has been striving to improve the portability of social health insurance through changing health insurance policy and constructing a national health insurance information platform. However, till now, the information platform among different provinces has not been fully interconnected.[33] Moreover, these efforts mainly focus on insurance reimbursement for cross-province inpatient care rather than outpatient care.

Another important finding of this study is that migrants who got reimbursement received larger reimbursement amounts if they migrated more broadly. Compared with migration within a city, migration across cities or provinces was significantly associated with about 30% higher

**Table 3** Association between insurance programmes, migration scope, other factors and social health insurance benefits

| Variables | Probit model | | Generalised linear model | | | |
| | Probability of getting reimbursements | | Reimbursement amount | | Reimbursement ratio | |
| | Coefficient | SE | Coefficient | SE | Coefficient | SE |
|---|---|---|---|---|---|---|
| Social health insurance (referred to NCMS) | | | | | | |
| URBMI | 0.021 | 0.049 | 0.222 | 0.136 | 0.147** | 0.031 |
| UEBMI | 0.375** | 0.044 | 0.428** | 0.094 | 0.201** | 0.023 |
| Age (years) | 0.010 | 0.012 | 0.026 | 0.030 | 0.006 | 0.006 |
| Age$^2$ (years) | 0.000 | 0.000 | 0.000 | 0.000 | 0.000 | 0.000 |
| Female | 0.024 | 0.031 | −0.004 | 0.078 | 0.018 | 0.016 |
| Married | 0.063 | 0.051 | −0.037 | 0.146 | 0.010 | 0.032 |
| Education attainment (referred to primary school and below) | | | | | | |
| Junior high school | 0.039 | 0.038 | 0.108 | 0.099 | 0.014 | 0.020 |
| Senior high school | 0.081 | 0.050 | 0.014 | 0.125 | 0.027 | 0.026 |
| College and above | −0.038 | 0.068 | −0.015 | 0.173 | 0.032 | 0.039 |
| Monthly household income per capita (referred to first quintile) | | | | | | |
| Second quintile | 0.062 | 0.048 | 0.328** | 0.124 | −0.029 | 0.026 |
| Third quintile | −0.033 | 0.051 | 0.014 | 0.139 | −0.041 | 0.028 |
| Fourth quintile | 0.047 | 0.049 | 0.179 | 0.124 | −0.031 | 0.026 |
| Fifth quintile | 0.029 | 0.053 | 0.336* | 0.137 | −0.023 | 0.030 |
| Having any job | −0.070 | 0.042 | −0.247* | 0.109 | 0.010 | 0.022 |
| Rural hukou | 0.088 | 0.053 | −0.106 | 0.126 | 0.000 | 0.031 |
| Living in an urban area | 0.063* | 0.032 | −0.012 | 0.088 | 0.004 | 0.017 |
| Geographical scope of migration (referred to migration across counties within a city) | | | | | | |
| Across cities within a province | −0.147** | 0.041 | 0.334** | 0.093 | −0.019 | 0.019 |
| Across provinces | −0.260** | 0.038 | 0.272** | 0.093 | −0.032 | 0.019 |
| Reasons for migration (referred to seeking jobs) | | | | | | |
| Family members following migrants | −0.007 | 0.050 | 0.553** | 0.137 | 0.047 | 0.027 |
| other reasons | −0.025 | 0.087 | −0.224 | 0.207 | 0.024 | 0.048 |
| Migration duration (referred to 0–) | | | | | | |
| 1– | −0.014 | 0.045 | 0.078 | 0.125 | −0.032 | 0.026 |
| 5– | 0.017 | 0.051 | 0.162 | 0.135 | −0.022 | 0.028 |
| 10+ | −0.011 | 0.052 | 0.085 | 0.140 | −0.040 | 0.029 |
| Facility level of hospitalisation (referred to primary care facility) | | | | | | |
| Secondary hospital | −0.031 | 0.048 | 0.603** | 0.139 | −0.046 | 0.032 |
| Tertiary hospital | 0.083 | 0.048 | 1.065** | 0.135 | −0.083** | 0.031 |
| Private hospital | −0.156* | 0.063 | 0.092 | 0.188 | −0.027 | 0.043 |
| Observations | 1165 | | 663 | | 663 | |

Note: ** and * denote statistical significance at 1% and 5% level, respectively.
NCMS, New Rural Cooperative Medical Scheme; UEBMI, Urban Employee Basic Medical Insurance; URBMI, Urban Resident Basic Medical Insurance.

reimbursement amount per inpatient stay. This is mainly attributed to higher healthcare costs in larger and more affluent regions where migrants preferred to locate. However, these reimbursements were paid primarily by health insurance funds in smaller counties/cities which were less affluent. This may pose financial difficulties for local governments in those out-migration regions if the current barriers to reimbursement are eliminated, and there will be more significant amounts of insurance funds flowing into the healthcare system in more prosperous in-migration regions, which will worsen the already skewed regional inequity in economic development and health. Therefore, the inverse subsidies of health insurance funds from the less-developed out-migration regions to the highly developed in-migration regions have become an ongoing challenge in countries with mass domestic

**Table 4** Association between migration scope and benefit of social health insurance programmes

| Variables | Probit model | | Generalised linear model | | | |
| | Probability of getting reimbursements | | Reimbursement amount | | Reimbursement ratio | |
| | Coefficient | SE | Coefficient | SE | Coefficient | SE |
|---|---|---|---|---|---|---|
| **NCMS subsample** | | | | | | |
| Geographical scope of migration (referred to migration across counties within a city) | | | | | | |
| Across cities within a province | −0.139** | 0.052 | 0.410** | 0.120 | −0.044 | 0.023 |
| Across provinces | −0.333** | 0.048 | 0.282* | 0.118 | −0.055* | 0.024 |
| **URBMI and UEBMI subsample** | | | | | | |
| Geographical scope of migration (referred to migration across counties within a city) | | | | | | |
| Across cities within a province | −0.135* | 0.061 | 0.339* | 0.154 | 0.073* | 0.033 |
| Across provinces | −0.118 | 0.061 | 0.285 | 0.152 | 0.039 | 0.033 |

Note: All models included confounding factors in table 3. ** and * denote statistical significance at 1% and 5% levels.
NCMS, New Rural Cooperative Medical Scheme; UEBMI, Urban Employee Basic Medical Insurance; URBMI, Urban Resident Basic Medical Insurance.

migration. Strategies that control health expenditures inflation and allow the central government to redistribute welfare funds to less-developed regions may be warranted to address the financial difficulties faced by out-migration regions.[6]

In addition to the above inequity related to migration, there are two other types of inequity in benefits due to the fragmented social insurance system and income inequity. Although China has almost achieved universal insurance coverage through social health insurance expansion,[6] migrants who enrolled in UEBMI and URBMI benefited more than those enrolled in NCMS.[10 34] Several national studies found that among the general population, UEBMI enrollees had a higher benefit level than those covered by URBMI or NCMS.[35 36] Even for the same condition—tuberculosis inpatient care—the reimbursement rate was the highest for UEBMI enrollees, followed by URBMI and NCMS enrollees in 2012.[34] Income inequity also explains some benefit inequities. Among the benefit recipients, we find that migrants with higher income received greater reimbursement amounts, which is consistent with previous studies among the general population.[16 19 37] On average, the higher-income group tended to have higher inpatient expenditure, and it was not surprising that they also received greater reimbursement amounts.[38]

This study contains some limitations. First, the outcomes related to reimbursement were self-reported, leading to measurement bias. Future research will use

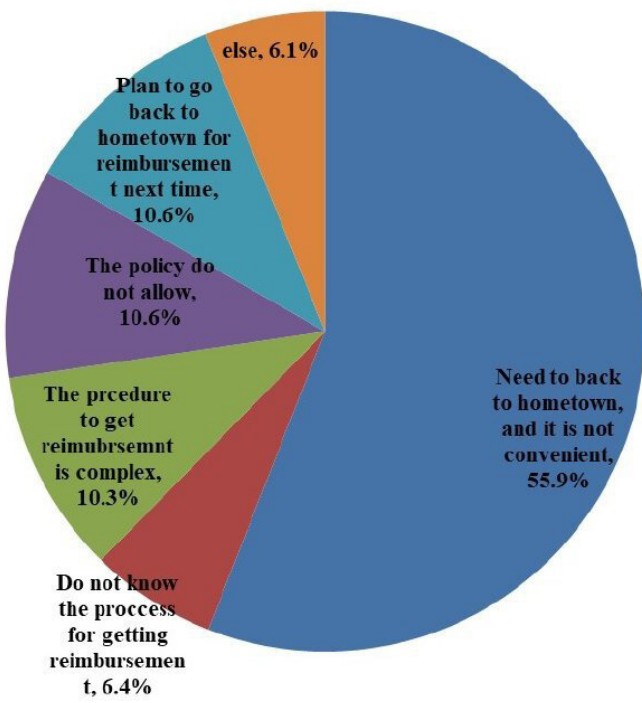

**Figure 1** Proportions of reasons not getting reimbursement from social health insurance (%).

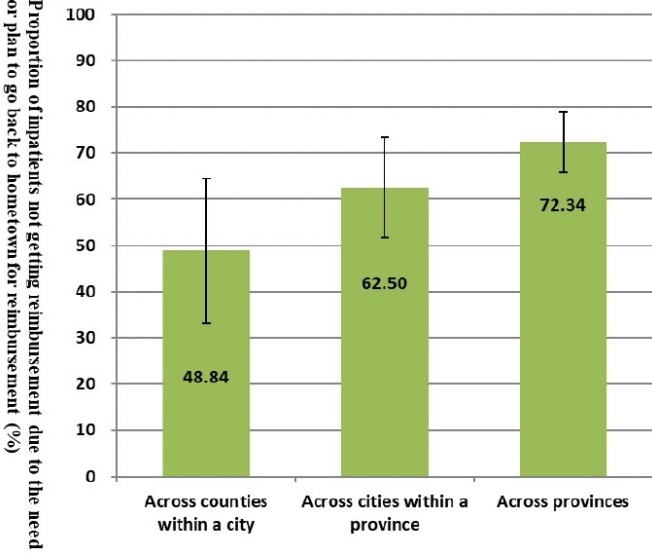

**Figure 2** Proportion of inpatients not getting reimbursement due to the need or plan to go back to hometown for reimbursement, by geographical scope of migration (%). Proportions and 95% CI are shown.

health insurance claims data to minimise this bias. Second, health status may influence healthcare utilisation and whether a person chooses to migrate. Studies have detected the healthy migrant effect for internal migrants in China, showing that healthier people were more likely to migrate and move farther away from home.[39] Unfortunately, there was no measurement of the health status of migrants in this dataset. Third, we can not accurately distinguish the administrative site of social health insurance, which may affect the insurance benefit for internal migrants. To reduce this bias, we conduct the analysis separately on two subsamples of NCMS enrollees and UEBMI and URBMI enrollees. Lastly, the 2014 China National Internal Migrant Dynamic Monitoring Survey is a dataset eight years ago, which may limit its implications for current issues.

## Conclusions

This study has important policy implications not only for China but also for other developing countries that experience rapid urbanisation and internal migration. In some low/middle-income countries (eg, Mexico), health insurance programmes are administrated locally and separated across regions,[40 41] hindering the insurance benefits when people migrate across regions. When their internal migrants move beyond the administrated region, they face the same problem as the Chinese migrants.

In China, the broader the migration scope, the lower the probability of receiving reimbursements from the social health insurance and the reimbursement ratio; but among those who received reimbursements, the broader the migration scope, the larger amounts they were reimbursed for healthcare use. This unequal benefit distribution may be attributed to the fragmented insurance design, which relies on localised administration and later reimbursement approach that migrating patients pay for health services up-front and get reimbursement later from health insurance. To improve the equity in social insurance benefits, China has been promoting the portability of social health insurance, immediate reimbursement for inpatient care used across regions, and a more integrated health insurance system. Efforts should also be made to control inflation of healthcare expenditures and prevent inverse government subsidies from out-migration regions to in-migration regions.

**Contributors** ZH designed the study and drafted the manuscript. HW conducted the literature review, analysed the data, and codrafted the manuscript. DL and DZ revised the manuscript. ZH is responsible for the overall content as guarantor. All authors read and approved the final manuscript being submitted.

**Funding** This work was supported by the National Natural Science Foundation of China (No. 71874034) and the National Key R&D Program of China (No. 2018YFC1312600 and 2018YFC1312604).

**Disclaimer** The funders had no role in study design, data collection and analysis, preparation of the paper, or the decision to publish.

**Competing interests** None declared.

**Patient and public involvement** Patients and/or the public were not involved in the design, or conduct, or reporting, or dissemination plans of this research.

**Patient consent for publication** Not applicable.

**Ethics approval** This study involves human participants but was not approved because the data used in this paper were publicly available, provided by the National Health and Family Planning Commission of China. Thus, there is no necessary to conduct an additional ethics approval.

**Provenance and peer review** Not commissioned; externally peer reviewed.

**Data availability statement** Data may be obtained from a third party and are not publicly available. The data used in this paper were provided by the National Health and Family Planning Commission of China. We signed a legally binding agreement with the Commission that we would not share the original data with any third parties. Although researchers who are interested can apply to get access to the data at: http://www.chinaldrk.org.cn and Email: ldrkzxsj@163.com, it is up to the National Health and Family Planning Commission to make the final decision.

**ORCID iDs**
Haiqin Wang http://orcid.org/0000-0002-0452-2442
Zhiyuan Hou http://orcid.org/0000-0003-3413-0076

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
