## [Reviewer comments · BMJ Open]

ARTICLE DETAILS

TITLE (PROVISIONAL)	How does domestic migration pose a challenge in achieving equitable social health insurance benefits in China? a National Cross-Sectional Study
AUTHORS	Wang, Haiqin; Liang, Di; Zhang, Donglan; Hou, Zhiyuan

VERSION 1 – REVIEW

REVIEWER	Nima Khodakarami Penn State Beaver
REVIEW RETURNED	10-Feb-2022

GENERAL COMMENTS	The paper is interesting and well-written. I am concerned that the implication of findings is limited to China only.
--

REVIEWER	Sebastian Salas-Vega London School of Economics and Political Science Health and Social Care
REVIEW RETURNED	14-Feb-2022

GENERAL COMMENTS	General Comments • Thank you for the opportunity to review this manuscript. This is in general a well-written study that offers an important perspective on the relationship between mass internal migration in China and social health insurance reimbursements and benefits. Specific Comments Introduction • Here and throughout manuscript, there were a few minor issues related to language that can be addressed should the manuscript be considered for publication. E.g. I would caution against using language like “the most” in line 44.• Otherwise, the introduction offered a robust description of the different insurance programs and their methods of financing• Most of the literature that the authors cite as part of their review on inequities in the distribution of social health insurance benefits is somewhat dated, e.g. citations #11-14. Could the authors update & expand on their review?• Also, some of the health insurance reforms that the authors discuss were implemented years ago. While the authors may be right that the literature examining the issue of inequities in “benefit distribution of social health insurance by scope of migration,” have there been any recent developments (e.g. policy or otherwise) that would make this analysis all the more timely? For example, is there growing recognition by those in Chinese academia or policy that there are “challenges for migrants to get insurance benefits under the current insurance policy design”? Methods / Results
--

	 • For their study, the authors use the 2014 China National Internal Migrant Dynamic Monitoring Survey, a dataset that is now 8 years out of date. Are there newer iterations of the survey that could be used instead? Are there other, newer data sources? I wouldn't say that this issue is alone cause for rejection, but it may leave the reader wondering how to potentially act on the insights generated by the study. Particularly if newer data aren't available, the authors should frame their discussion of findings around their review of relevant policies and recent phenomena regarding mass internal migration flows over the intervening 8 years – e.g. how should these affect our interpretation of results? • The authors base their analysis around the notion of the “scope of migration” of migrants within China, comparing for instance results for those who migrated across cities and provinces versus those who migrate within a city. One example of these results is that those who migrated across provinces had the lowest reimbursement ratio. These are interesting results that both international & Chinese readers would be appreciate reading. However, to make these results easier to interpret, might the authors be able to provide a bit more context into the type of individuals who might travel further vs shorter distances in China? How might these subgroups be characterized (e.g. are those that travel further distances older, more likely to be female, more highly educated, poorer)? Table 1 provides the characteristics of the study sample (total, NCMS, URBMI & UEBMI), but not by the scope of migration subgroups. Likewise, would the authors be able to explain why the geographic scope of migration was categorized using the three subgroups that were considered in the paper? Discussion  • Thank you for including a discussion of study limitations. Among these, the authors acknowledge that e.g. reimbursements were self-reported, they lacked information on the health status of migrants, and that they “could not accurately distinguish the administration location of social health insurance.” • Could the authors clarify what they mean by “the administration location of social health insurance?” I believe you may be referring to administrative sites belonging to social health insurance programs that are tasked with administering benefits, e.g. processing reimbursements to patients. Is that correct? Apologies if I misunderstood.
--	--

VERSION 1 – AUTHOR RESPONSE

Response to Reviewer 1 Dr. Nima Khodakarami, Penn State Beaver

The paper is interesting and well-written. I am concerned that the implication of findings is limited to China only.

Responses: This study may have policy implications not only for China, but also for other developing countries with similar contexts. In some developing countries (e.g., Mexico), health insurance programs are administrated locally and separated across regions⁴⁰⁻⁴¹, hindering the insurance benefits when people migrated across regions. Internal migrants often move beyond the administrated region for work, and then would face the same problem as the Chinese when seeking health services.

In the conclusions (Line 469- 474, Page 18) , we added, “In some developing countries (e.g., Mexico), health insurance programs are administrated locally and separated across regions⁴⁰⁻⁴¹, hindering the insurance benefits when people migrate across regions. Internal migrants often move beyond the administrated region for work and then would face the same problem as the Chinese when seeking health services. ”

Response to Reviewer 2 Dr. Sebastian Salas-Vega, London School of Economics and Political Science Health and Social Care

Thank you for the opportunity to review this manuscript. This is in general a well-written study that offers an important perspective on the relationship between mass internal migration in China and social health insurance reimbursements and benefits.

1. Here and throughout manuscript, there were a few minor issues related to language that can be addressed should the manuscript be considered for publication. E.g. I would caution against using language like “the most” in line 44.

Response: We have revised this expression and look throughout the manuscript for language errors.

2. Otherwise, the introduction offered a robust description of the different insurance programs and their methods of financing.

Response: Thanks for your comment.

3. Most of the literature that the authors cite as part of their review on inequities in the distribution of social health insurance benefits is somewhat dated, e.g. citations #11-14. Could the authors update & expand on their review?

Response: We have updated the literature and added the following nine latest studies.

The following review was added (Line 110-122, Page 5): “Evidence from URBMI (between 2007 and 2011)¹⁹ and the social health insurance (between 2014 to 2016)²⁰ revealed that the lower-income groups benefited less than the higher-income groups. The poorest groups within URBMI and NCMS were consistently more likely to forego hospitalization services recommended by doctors than their wealthier counterparts between 2008 to 2018⁹ In recent years, more literature documented barriers to preventing migrants from gaining access to health care²¹⁻²² and health insurance had little influence on healthcare utilization among internal migrants²²⁻²⁴. Li’s study further concluded that NCMS did not play a significant role in reducing out-of-pocket payments for elderly migrants between 2005 to 2014²⁵.”

Reference:

9 X Yan, Y Liu, M Cai, et al. Trends in disparities in healthcare utilisation between and within health insurances in China between 2008 and 2018: a repeated cross-sectional study. *Int J Equity Health* 2022;21(1).

19 J Pan, S Tian, Q Zhou, W Han. Benefit distribution of social health insurance: evidence from china’s urban resident basic medical insurance. *Health Policy Plann* 2016;31(7):853-59.

20 L Diao, Y Liu. Inequity under equality: research on the benefits equity of Chinese basic medical insurance. *Bmc Health Serv Res* 2020;20(1).

21 X Ke, L Zhang, Z Li, W Tang. Inequality in health service utilization among migrant and local children: a cross-sectional survey of children aged 0–14 years in Shenzhen, China. *Bmc Public Health* 2020;20(1).

22 F Zhang, X Shi, Y Zhou. The Impact of Health Insurance on Healthcare Utilization by Migrant Workers in China. *Int J Env Res Pub He* 2020;17(6):1852.

23 H Wang, D Zhang, Z Hou, F Yan, Z Hou. Association between social health insurance and choice of hospitals among internal migrants in China: a national cross-sectional study. *Bmj Open* 2018;8(2):e18440.

24 X Qin, J Pan, GG Liu. Does participating in health insurance benefit the migrant workers in China? An empirical investigation. *China Econ Rev* 2014;30:263-78.

25 J Li, Y Huang, S Nicholas, J Wang. China’s New Cooperative Medical Scheme’s Impact on the Medical Expenses of Elderly Rural Migrants. *Int J Env Res Pub He* 2019;16(24):4953.

4. Also, some of the health insurance reforms that the authors discuss were implemented years ago. While the authors may be right that the literature examining the issue of inequities in “benefit distribution of social health insurance by scope of migration,” have there been any recent

developments (e.g. policy or otherwise) that would make this analysis all the more timely? For example, is there growing recognition by those in Chinese academia or policy that there are “challenges for migrants to get insurance benefits under the current insurance policy design”?

Response: We have added the new policy development. Although China started to address insurance reimbursements for migrants, there are still challenges for migrants to get insurance benefits under the current insurance policy design.

we added (Line 100-106, Page 5): “To address the insurance reimbursement issue for migrants, China promoted immediate reimbursement for cross-province medical care in 2014¹³. However, the immediate reimbursement policy was only used for inpatient services, and immediate reimbursement for outpatient services started a pilot in a few regions in 2020¹⁴. Despite these policies, the separation between the location of health care use and health insurance coverage would still be barriers to equity benefits¹⁵.”

Reference:

13 General Office of the State Council. Notice on key missions for deepening health system reform in 2014. In, 2014.

14 National Healthcare Security Administration. Notice on Promoting the Pilot Program of Cross-Provincial Direct Settlement of Outpatient Expenses. In, 2020.

15 S Chen, Y Chen, Z Feng, et al. Barriers of effective health insurance coverage for rural-to-urban migrant workers in China: a systematic review and policy gap analysis. *Bmc Public Health* 2020;20(1).

5. For their study, the authors use the 2014 China National Internal Migrant Dynamic Monitoring Survey, a dataset that is now 8 years out of date. Are there newer iterations of the survey that could be used instead? Are there other, newer data sources? I wouldn't say that this issue is alone cause for rejection, but it may leave the reader wondering how to potentially act on the insights generated by the study. Particularly if newer data aren't available, the authors should frame their discussion of findings around their review of relevant policies and recent phenomena regarding mass internal migration flows over the intervening 8 years – e.g. how should these affect our interpretation of results?

Response: Thanks for your suggestion. Unfortunately, there is no new data sources available. We then further framed our discussion of findings with the recent policy development.

We added the following in the discussion (Line 416-422, Page 17): “To resolve this inequity, China has made efforts to improve the portability of social health insurance through policy development and the building of a national health insurance information platform. However, till now, the information platform among different provinces has not been fully interconnected³³. Moreover, these efforts mainly focus on insurance reimbursement for cross-province inpatient care rather than outpatient care.”

Reference:

33 Y Chen. Study On The Optimization of Long-Distance Medical Insurance of Public Hospitals In Shanghai: Based on Holistic Governance Theory. Shanghai Jiao Tong University, 2020.(Chinese)

6. The authors base their analysis around the notion of the “scope of migration” of migrants within China, comparing for instance results for those who migrated across cities and provinces versus those who migrate within a city. One example of these results is that those who migrated across provinces had the lowest reimbursement ratio. These are interesting results that both international & Chinese readers would be appreciate reading. However, to make these results easier to interpret, might the authors be able to provide a bit more context into the type of individuals who might travel further vs shorter distances in China? How might these subgroups be characterized (e.g. are those that travel further distances older, more likely to be female, more highly educated, poorer)? Table 1 provides the characteristics of the study sample (total, NCMS, URBMI & UEBMI), but not by the scope of migration subgroups. Likewise, would the authors be able to explain why the geographic scope of migration was categorized using the three subgroups that were considered in the paper?

Response: Since the social health insurance programs are administrated and financed by county or city, we hypothesized there are benefit disparities for migrants rooted in the separate administration of

insurance programs. Therefore, our study categorized migrants according to their scope of migration, which can capture the degree of separate administration of insurance programs. We added the above content (Line 176-179, Page 7) to the “Measurements”

We now compared the characteristics of the study sample by the “scope of migration” of migrants, and added Appendix Table 1. We found that the demographic and socio-economic characteristics were similar among three scopes of migration subgroups, including age, sex, marriage status, education, job status, Hukou status, reasons for migration, migration duration, and health insurance programs they enrolled in. The only exception was income and living areas, and migrants across provinces had more income and were less likely to live in urban areas than other groups. Migrants usually choose to move broader due to higher return on income. Due to the similar characteristics across three scope of migration subgroups, insurance benefit disparities can be explained by the separate administration of insurance programs.

We have added the above contents in the revised version (Line 279-285, Page11).

Appendix Table 1 Characteristics of the study sample by migration scope

	Across counties within a city	Across cities within a province	Across provinces	F/ χ^2	P value
Age (Years)	38.41(\pm 10.35)	37.99(\pm 9.62)	37.01 (\pm 9.47)	2.248	0.106
Female	137(46.28%)	154(46.0%)	241(45.13%)	0.12	0.942
Married	262(88.51%)	298(88.96%)	480(89.89%)	0.424	0.809
Education attainment	6.979	0.323			
Primary school and below	72(24.32%)	78(23.28%)	129(24.16%)		
Junior high school	137(46.28%)	154(45.97%)	263(49.25%)		
Senior high school	64(21.62%)	62(18.51%)	86(16.10%)		
College and above	23(7.77%)	41(12.24%)	56(10.49%)		
Monthly household income per capita (Yuan)	1830.88(\pm 1272.07)	1968.94(\pm 1589.62)	2672.67(\pm 2600.04)	20.506	0.000
Having any job	226(76.35%)	264(78.81%)	437(81.84%)	3.692	0.158
Rural Hukou	268(90.54%)	293(87.46%)	462(86.52%)	2.934	0.231
Living in urban area	237(80.07%)	241(71.94%)	330(61.80%)	31.386	0.000
Reasons of migration	8.13	0.087			
Seeking jobs	242(81.76%)	273(81.49%)	466(87.27%)		
Family members following migrants	44(14.86%)	51(15.22%)	51(9.55%)		
Other reasons	10(3.38%)	11(3.28%)	17(3.18%)		
Migration duration (Years)	10.475	0.106			
0~1	43(14.53%)	32(9.55%)	72(13.48%)		
1~5	135(45.61%)	135(40.30%)	214(40.07)		
5~10	65(21.96%)	80(23.88%)	127(23.78%)		
10 and above	53(17.91%)	88(26.27%)	121(22.66%)		
Social health insurance programs	8.376	0.079			
NCMS	216(72.97%)	214(63.88%)	347(64.98%)		
URBMI	28(9.50%)	39(11.64%)	53(9.93%)		
UEBMI	52(17.57%)	82(24.48%)	134(25.09%)		

Note: * mean (\pm Standard Deviation).

7. Could the authors clarify what they mean by “the administration location of social health insurance?” I believe you may be referring to administrative sites belonging to social health insurance programs that are tasked with administering benefits, e.g. processing reimbursements to patients. Is that correct? Apologies if I misunderstood.

Response: Yes, you are right. We revised the terms for ease of understanding. (Line 462-463, Page 18)